# MicroRNA-371a-3p—The Novel Serum Biomarker in Testicular Germ Cell Tumors

**DOI:** 10.3390/cancers15153944

**Published:** 2023-08-03

**Authors:** Tim Nestler, Justine Schoch, Gazanfer Belge, Klaus-Peter Dieckmann

**Affiliations:** 1Department of Urology, Federal Armed Forces Hospital Koblenz, 56072 Koblenz, Germany; 2Department of Tumour Genetics, University Bremen, 28359 Bremen, Germany; 3Department of Urology, Asklepios Klinik Altona, 22763 Hamburg, Germany

**Keywords:** MicroRNA, miRNA, biomarker, testicular cancer, germ cell tumor

## Abstract

**Simple Summary:**

This review describes that miR371 is much more sensitive and specific than the classical serum biomarkers of testicular germ cell cancer. There is a huge potential for miR371 for initial diagnosis, monitoring treatment, and follow-up. However, miR371 is not expressed in teratoma. Currently, studies are underway to determine the importance of miR371 in the early detection of recurrences of testicular tumors under surveillance. Further studies will examine the role of miR371 in residual tumors after chemotherapy, with the aim of uncovering viable tumor components.

**Abstract:**

Introduction: Testicular germ cell tumors (TGCTs) are a paradigm for the use of serum tumor markers in clinical management. However, conventional markers such as alpha-fetoprotein (AFP), beta-human chorionic gonadotropin (hCG), and lactate dehydrogenase (LDH) have quite limited sensitivities and specificities. Within the last decade, the microRNA-371a-3p (miR371) emerged as a possible new biomarker with promising features. Areas covered: This review covers the typical features as well as possible clinical applications of miR371 in TGCT patients, such as initial diagnosis, therapy monitoring, and follow-up. Additionally, technical issues are discussed. Expert opinion: With a sensitivity of around 90% and specificity >90%, miR371 clearly outperforms the classical serum tumor markers in TGCTs. The unique features of the test involve the potential of modifying recent standards of care in TGCT. In particular, miR371 is expected to aid clinical decision-making in scenarios such as discriminating small testicular TGCT masses from benign ones prior to surgery, assessing equivocal lymphadenopathies, and monitoring chemotherapy results. Likewise, it is expected to make follow-up easier by reducing the intensity of examinations and by sparing imaging procedures. Overall, the data presently available are promising, but further prospective studies are required before the test can be implemented in standard clinical care.

## 1. Introduction

Testicular germ cell tumors (TGCTs) represent the most common malignancy in males aged 20–40 years, with incidences of 7–10 patients per 100,000 men per year in western countries [1]. The incidence of TGCTs increased over the past 30 years, while the mortality rates decreased due to improvements in clinical management [2,3]. Today, more than 90% of all TGCT patients can be cured [4]. Thus, current clinical research strategies focus on optimizing therapeutic intensity with the aim of reducing long-term toxicity of treatment, thus preserving quality of life. TGCTs are clinically subdivided into two major histological subgroups: pure seminoma, accounting 60% of TGCTs, and 40% non-seminomas (NSGCT), encompassing all other histological types and their various combinations according to the 2022 WHO classification [5]. Decision-making regarding treatment of TGCTs is firstly based on the major histologic subtype and secondly on clinical staging according to cross-sectional imaging and measurement of the serum tumor markers (STMs) [6]. Traditionally, there are three STMs, all of which represent proteins, chemically: alpha-fetoprotein (AFP), beta-human chorionic gonadotropin (hCG), and lactate dehydrogenase (LDH). However, the main limitation of these markers is their low sensitivity and specificity for tumor detection, since they are quite differently expressed in the various TGCT subtypes and tumor stages [7]. In pure seminomas, hCG and LDH are elevated in only 28%, and 29%, respectively, while AFP is negative by default. In NSGCT, hCG, AFP, and LDH are elevated in 53%, 60%, and 39%, respectively [8].

To overcome the limitation of the low sensitivity of the established STMs in TGCT, a number of novel markers were investigated during the last decades, such as neuron-specific enolase, placental alkaline phosphatase, TRA 1-60, and others more [9,10,11]. However, none could ever qualify for clinical implementation [12]. During the last decade, research mainly focused on microRNAs (miRs), which, in contrast to the protein-based traditional STMs, represent small noncoding RNA molecules consisting of around 22 nucleotides. The physiological function of miRs involves the posttranscriptional regulation of gene expression and RNA silencing, representing an epigenetic modification of protein biosynthesis, but may also be part of carcinogenesis in terms of oncogenes or tumor suppressor genes [13,14]. There is wide-spread consensus that blood-based biomarkers should ideally fulfil the seven criteria established by Lange and Winfield in 1987, which are shown in Table 1 [15].

The traditional tumor markers of TGCT fulfil only part of these classical criteria. Particularly, the first condition is not matched by the three markers. LDH is rather a marker of cell death and is elevated in serum in many other diseases with high cell turnover. AFP and hCG are produced in several diseases other than TGCT, and serum levels may be influenced by endocrinological and metabolic disturbances [16,17]. By contrast, the novel miR-based markers appear to fulfil all of the conditions.

## 2. Body

### 2.1. Rationale for Investigating microRNAs and First Clinical Studies

A lot of evidence suggests that TGCTs mimic embryonic development to a certain extent, retaining the molecular biological and biochemical features of embryonic stem cells (ESCs). ESCs represent continuously growing stem cells, which are characterized by pluripotency and self-renewal [18]. Importantly, ESCs express various specific miRNA-clusters, particularly miR-371-3 and miR-302/367, that play key roles in regulating pluripotency and differentiation. These miRs are human ESC-specific, and their expression is downregulated during cell development [19,20]. Embryonic miRs appeared to be promising molecular biomarkers in TGCT for cancer detection because of their proven stability in circulating body fluids [21]. Accordingly, these miR-clusters were detected in TGCT tissue during the first decade of this century [22]. Murray et al. were the first to document the measurable levels of these microRNAs in serum [23]. The presence of miRs-371-3 in the serum of TGCT patients was confirmed in a small series of 11 patients, and it was also shown that miR levels drop to normal after orchiectomy [24]. As the correlation of serum levels with tumor bulk is one of the prerequisites of a valuable tumor marker [15], the embryonic microRNAs became candidates for the next generation tumor marker of TGCTs. During 2012–2018, several independent studies showed that miR-371a-3p (miR371) involves an excellent receiver operating characteristic (area under curve [AUC]: 0.93–0.95), resulting in a sensitivity of 84–89% and specificity of 90–99% for TGCT, outperforming other candidate miRs, such as miR-372 (AUC: 0.79–< 0.9), miR-373 (AUC: 0.77–< 0.9), and miR-367 (AUC: 0.82–< 0.9) [25,26,27,28]. From these studies, miR371 emerged as the most promising one, with regard to clinical applications. Consequently, further studies only focused on this single miR. An important feature of miR371 is that contrary to traditional markers, miR371 serum levels are elevated in both seminoma and nonseminoma. However, a major drawback is the non-expression of miR371 in the subtype of teratoma and only very weak expression in germ cell neoplasia in situ [29]. Currently, there is a large body of evidence suggesting the substantial utility of miR371 in TGCT patients in various clinical settings, such as diagnosing small testicular masses, initial staging of TGCT, monitoring of treatment response, evaluation of residual masses after chemotherapy, and follow-up of TGCT patients.

### 2.2. Non-Specific Small Testicular Masses

With the refinements of scrotal ultrasound technique, incidentally detected small testicular neoplasms represent a clinical scenario of increasing frequency (Figure 1) [30]. Testicular masses with elevated STMs are very likely to represent TGCT. However, the majority of small testicular masses involve non-specific and benign histologies other than TGCT, e.g., atrophic or fibrotic lesions, Sertoli or Leydig cell tumors, or Leydig’s cells hyperplasia, and thus present without elevated STMs [31]. In patients with suspected non-TGCT of small size (<1 cm/3 cm^3^), testis-sparing surgery with intraoperative frozen section examination is considered to be the standard of care [6,32,33]. If traditional STMs are negative, miR371 might be of value for differentiating between malignant and benign masses prior to surgery in order to reduce the overtreatment in patients with benign tumors. In this pre-orchiectomy setting, serum levels of miR371 were prospectively analyzed [34] with regard to the sensitivity of miR371 in subgroups of TGCT patients with different tumor sizes. In NSGCT, there were no differences in sensitivity among various tumor size categories with a sensitivity of 100%, even in lesions < 10 mm. However, in seminomas, there was a significantly decreasing sensitivity of miR371 with decreasing tumor size categories with expression rates of 59% and 77% in lesions of <10 mm and 10–20 mm, respectively. Because of the study design, which included only patients with proven TGCT and healthy controls, no statement about the specificity of miR371 in identifying small TGCT could be made. However, there was a specificity of 94.0% for the healthy control group. A recent study retrospectively investigated 641 patients with testicular neoplasms, including 87 patients with subcentimeter testicular masses [35]. Within the group of small masses (<10 mm), 50.6% were benign tumors and miR371 was elevated in 44.1% of these patients compared to AFP, hCG, or LDH with 2.3%, 11.6%, and 7.0%, respectively. As tumor size is usually a subgroup analysis in larger study populations, further prospective studies of small STM-negative testicular masses including miR371 are needed to obtain sufficient data about the sensitivity and specificity of miR371 in these small tumors.

In practical terms, all patients with small testicular masses should undergo histological confirmation regardless of their STM or miR371 level because especially small seminomas might lack expression of miR371 in more than 50% of cases. One future vision could be that miR371-positive patients may directly undergo orchiectomy without intraoperative frozen section examination.

### 2.3. Use of miR371 in Primary Diagnosis of TGCT

To date, twelve studies investigated the utility of miR371 for the primary diagnosis of TGCT, encompassing a total of 1632 TGCT patients (Table 2) [25,26,27,28,34,36,37,38,39,40,41,42]. The diagnostic accuracy was reported in seven studies and resulted in a very large area under the receiver operating characteristics curve (AUC) of ≥0.93. Six studies reported a sensitivity of 84.7–96%, and a specificity of 84.7–100% [25,27,28,34,37,39] (Table 2). The outstanding performance characteristics of miR371 are clearly superior to the other embryonic miRs, and even panels of combined miRs do not yield better results.

So far, there are four prospective studies. The largest prospective multicenter study was conducted by the German Testicular Cancer Study Group and included 616 TGCT patients and 258 controls [34]. For the group of non-metastatic patients, the authors reported an AUC of 0.953, while metastatic patients even arrived at 0.996 (Figure 2). Nappi et al. evaluated miR371 in plasma of 110 patients with TGCT of 46 with active disease [37]. All miR371-positive patients had TGCT (specificity: 100%) with two (4%) false-negative patients (sensitivity: 96%). A small Danish study was prospective too, but it focused mainly on technical aspects, such as miRNA isolation and testing of different miRs [38], and will not be discussed further here. Myklebust et al. evaluated miR371 by using conventional real-time quantitative PCR (RT-qPCR) and compared it to digital droplet PCR (RT-ddPCR) [43]. All previous studies used RT-qPCR. Usually, a pre-amplification is necessary because miR levels are approaching the limits for detection by RT-qPCR. The main advantage of RT-dd-PCR is that pre-amplification is not necessary, which results in a time advantage and thus higher throughput. Since individual PCR reactions take place simultaneously in thousands of droplets in RT-ddPCR, specificity and sensitivity are usually slightly increased compared to conventional RT-qPCR. In their analyses, both technics (RT-ddPCR and RT-qPCR) performed in a similar fashion. Out of the 180 samples (159 proven T GCT samples and 21 controls) analyzed with RT-ddPCR, all miR371-positive patients had TGCT (specificity: 100%), with 17 (10.7%) false-negative patients (sensitivity: 89%). Another study examining ddPCR was published in 2022 [42]. Here, plasma was obtained from 82 patients before orchiectomy, of which 31 had TGCT. A sensitivity of 93.6% and specificity of 100% was reported for this cohort, resulting in an AUC of 0.98.

In aggregate, the studies available to date show excellent specificities, with somewhat lower sensitivities for the primary diagnosis of TGCT, which might relate to the decreasing sensitivity of miR371 in lesions smaller than 2 cm, particularly in seminoma. For clinical practice, this could imply that all miR371-positive patients should be referred for orchiectomy, and even in the case of inconclusive sonography, frozen section examination would not be necessary. Due to the lower sensitivity, and thus the risk of false-negative miR371 results, all conspicuous testicular masses detected sonographically should undergo histological examination by inguinal exposure and excision.

### 2.4. Use of miR371 in Management and Follow-Up of Early Stage TGCT

Clinically, it is of particular interest to identify patients with radiographically invisible micro-metastatic disease (clinical stage I = cSI) right after orchiectomy, since occult metastases will inevitably progress. Currently, clinical evidence for occult metastases is derived from rather crude morphological tools such as lymphovascular invasion (LVI) in NSGCT and tumor size in seminoma. Patients with LVI_pos_ involve a 47.5% risk of progression, opposed to still 16.9% in LVI_neg_ patients. In seminomas with tumor size > 4 cm, the risk of relapse within 5 years is 17.4–27.0%, while it is 4.5–13.4% in smaller tumors [44,45]. As these traditional diagnostic tools are rather unspecific and only a little sensitive, with a number of patients relapsing in the absence of these markers, the advent of miRNAs raised the hope for establishing a more informative marker for identifying patients destined to progress. Particularly, miR371 was deemed to be helpful in distinguishing occult metastatic from non-metastatic disease. Accordingly, Bagrodia et al. investigated the impact of miR371 on making the correct treatment decisions in NSGCT cSIA/B patients [46]. They reported that with miR371 measurements, appropriate treatment decisions significantly increased from 65% to 94% in cSIA patients’, and from 50% to 92% in cSIB.

Basically, serum miR371 levels correlate with the tumor bulk of TGCT, which was concluded from significant increases in miR levels with clinical stages [34]. Likewise, miR371 levels significantly decrease after orchiectomy in cSI patients, resulting from the excision of the primary tumor. Postoperatively, serum miR371 levels dropped in 91.8% of 424 patients, but 20% of metastasized cases remained somewhat elevated. The reason for the inadequate drop of these cSI patients remained elusive in that study because no follow-up of these patients was available (Figure 3). However, it was hypothesized that incomplete decreases could denote occult metastasis. This hope was originally nourished by Lafin and colleagues, who showed that miR371 was elevated in all cases with metastatic disease among 24 patients with NSGCT cSI-IIA/B disease undergoing primary retroperitoneal lymph node dissection [47]. Conversely, all cases harboring teratoma or benign masses were miR371-negative, except for one patient with a benign finding, resulting in a sensitivity of 100% and a specificity of 92%. However, a study from Toronto refuted the appealing hypothesis in a follow-up trial on 151 cSI patients, of whom 23% relapsed [48]. It is noteworthy that patients going to relapse in the later course did not have elevated levels right after orchiectomy. Conflicting data were reported recently from the Vancouver group. These authors noted a significantly shorter relapse-free survival time in patients with postoperatively elevated miR levels [49]. On the other hand, in another prospective study, it was shown that postoperative miR371 levels were not associated with the known risk factor for progression (LVI_pos_) in NSGCT cSI patients [50]. However, a number of methodological issues were raised with these reports. Thus, there is currently no unequivocal evidence for an association between postoperatively elevated miR371 levels and future progressive disease in cSI patients [50]. Future studies with clearly defined time points of blood aspiration for miR detection have to be awaited. 

Follow-up examinations in cSI patients after completion of treatment usually include clinical examination and tumor marker monitoring in addition to imaging [6]. Computed tomography involves ionizing radiation, and the cumulative dosage following several exposures may cause a predisposed increased risk of secondary malignancies [51]. In addition, imaging technology consumes a lot of energy with each examination, and is therefore quite expensive and not very sustainable. Therefore, it appeared rational to include miR371 in follow-up schedules, and the hope was to reduce the frequency of cross-sectional imaging and also to save costs [52]. Table 3 summarizes the currently available data regarding the utility of miR371 for detection of relapses [34,48,49,53,54].

In a retrospective analysis based on 151 patients with cSI followed by surveillance, a total of 23% relapsed [48]. In relapsing patients, miR371 was elevated in 94.1% at the time of relapse compared to classical STM AFP and hCG, which were normal in 62%. This study also confirmed that levels of miR371 significantly correlate with metastatic extent (N0/1 vs. N2-3/M0 *p* = 0.05). In a prospective multicenter study, 38 of 46 relapsing patients had elevated miR371 levels (82.6%), and the median serum level of these patients was significantly higher than that of controls (*p* < 0.001), resulting in a sensitivity of 82.6%, a specificity of 96.1%, and an AUC of 0.921 for relapse detection [34]. It is noteworthy that miR371 levels decreased significantly in 28 of 29 patients (*p* < 0.001) after cure from relapses. Fankhauser et al. recently reported a series of prospectively collected serum samples of 33 TGCT patients with cSI disease during active surveillance without any adjuvant therapy [54]. A total of ten patients relapsed within a median of 7 months after orchiectomy and elevated miR371 levels were detected in all of them at relapse. For comparison, patients without recurrence had non-elevated miR371 levels, except for one patient with an elevation at one single timepoint. Of clinical importance, elevated miR371 levels were found to be elevated at a median of 2 months (range 0–5) ahead of detection of recurrences with standard follow-up examinations.

Terbuch et al. found miR371 levels of ten patients [53] to be significantly increased at relapse by comparison to miR levels from the same patients measured at the time without active disease (*p* = 0.014).

Preliminary data from the Vancouver group confirmed the high specificity of miR371 in detecting relapses in GCT cSI patients; however, the sensitivity was only 62.8% in that ongoing trial [49].

In all, the studies listed here provided promising data nourishing the view that miR371 can become a powerful diagnostic tool in the follow-up of TGCT patients that may detect recurrences even earlier than the traditional combination of imaging and measuring conventional STMs. However, prospective studies on sufficiently large populations are still pending.

### 2.5. Monitoring Chemotherapy with miR371

Measuring STMs prior to each cycle of chemotherapy is mandatory according to current guidelines [32,55]. As miR371 involves an excessively short half-life of less than 24 h [56], the measurement of this miR could be a powerful marker for rapidly identifying those patients who respond insufficiently to chemotherapy and who would require early intensification of therapy.

A multicentric prospective study reported results of repeat measurements of miR371 levels in 70 patients with cSIIA/B and 46 patients with cSIII during standard chemotherapy [34]. In cSIIA/B patients, miR371 levels significantly dropped after the first cycle of chemotherapy but did not further decrease thereafter. In cSIII patients, no further decrease was observed after the second cycle (Figure 4). Accordingly, two patients with fatal outcomes were shown to have increasing miR371 levels during chemotherapy. Likewise, Rosas Plaza et al. reported that miR371 levels decreased within the first week of chemotherapy in patients attaining complete response [57]. Mego et al. analyzed miR371 prior to chemotherapy in 180 patients [58]. In patients with elevated miR371 before chemotherapy, they found progression-free survival and overall survival to be inferior to those with normal miR levels. Equally, two other studies confirmed the significant association of progression-free survival with the median miR-serum levels before chemotherapy [53,57].

The velocity of the tumor marker decline was shown to prognosticate the chance of cure in patients with poor-prognosis GCT undergoing cisplatin-based chemotherapy [59,60]. It is rational to assume this association to be relevant also with the novel marker miR371. Moreover, as this marker does obviously decline much more rapidly than bHCG and AFP, it may be hypothesized that miR371 decline might represent an even better tool for predicting cure rates in poor-risk GCT patients.

In summary, the studies outlined here provide clear evidence for a correlation between miR371 decrease and therapy response, and there is initial evidence for the prognostic significance of elevated miR371 levels. However, the patient samples studied so far are still too small to justify a soon inclusion of the test in clinical practice. Of note, there is no data so far about the miR371 level kinetics during radiation therapy. This kind of study needs to be conducted soon.

### 2.6. Use of miR371 for Assessing Residual Masses after Chemotherapy

Approximately 70% of metastatic NSGCT patients who underwent first-line cisplatin-based chemotherapy attain complete STM normalization and are radiographically disease-free, corresponding to a complete remission (CR) [6,61]. In NSGCT patients with residual masses > 1 cm, postchemotherapy retroperitoneal lymph node dissection (pcRPLND) is required to resect viable TGCT and teratoma, which are present in approximately 10% and 40% of residual masses, respectively [6,62]. Conversely, necrosis is present in the pcRPLND specimen in about 50% of all patients undergoing pcRPLND. Principally, this large group of patients with necrosis only could be spared surgery if the presence of viable TGCT and teratoma could be clearly identified or ruled out prior to surgery. Accurate identification of cases requiring pcRPLND is of particular relevance, as this procedure represents major surgery with the potential of significant immediate or long-term morbidity. The rationale for surgical resection of residual masses relates to the resistance to standard chemotherapy or radiation of growing teratoma and somatic malignancy, as well as residual vital germ cell cancer. By today, neither clinical models nor modern imaging interpretation approaches such as radiomics were informative enough to accurately differentiate between necrosis and teratoma/viable TGCT prior to pcRPLND [63,64].

The Toronto group investigated miR371 levels of 82 NSGCT patients prior to pcRPLND [65]. They reported an AUC of 0.874 for the detection of vital TGCT and showed a sensitivity of 100% and a negative predictive value of 100% for retroperitoneal masses ≤ 3 cm. However, it should be noted that in this study, the calculations of performance characteristics were based on the comparison of viable TGCT vs. teratoma and necrosis, although the clinically relevant evaluation would be the comparison of viable TGCT and/or teratoma with necrosis (Figure 5). The choice of groups is explained by the negativity of miR371 for teratomas as previously described [34]. Likewise, Rosas Plaza et al. reported that all 44 patients with fibrosis/necrosis or teratoma upon pcRPLND were miR371-negative prior to surgery, while three of four patients with viable TGCT in their resected specimen were miR371-positive [57].

To close this “teratoma gap”, several biomarker studies were performed. Myklebust et al. performed miRNA next-generation sequencing on teratoma tissue. However, the validation of candidate markers in patients’ serum was unsuccessful [66], and miR375 was found to be upregulated in tissue of teratoma [67], indicating that this miR could be an additional valuable marker for evaluating residual nonseminomatous masses after chemotherapy [68].

Accordingly, the utility of miR375 in serum was examined by several groups, but miR375 levels in serum did not correlate with the presence of teratoma [69,70,71]. Additionally, Lafin et al. performed a comprehensive small RNA sequencing analysis, but a specific miRNA for teratoma could not be identified [69]. Lobo and colleagues suggested using a combination of circulating miR-371a-3p and the cell-free DNA of the hypermethylated promoter domain of the *RASSF1A* gene in serum to detect TGCTs including teratoma, and they reported a sensitivity of 100% [72]. However, these findings await validation in further independent studies. In conclusion, the probability of identifying specific serum biomarkers for teratoma seems to be limited. Therefore, a comprehensive analysis of mRNA and proteins on pcRPLND tissue was performed recently in order to identify a stable protein marker being expressed in teratoma tissue compared to necrosis [73]. Here, AGR2 and KRT19 were identified. These proteins were validated on an independent patient cohort, showing a significant overexpression in teratoma compared to necrosis (*p* < 0.0001 and AUC 1.0; sensitivity and specificity of 100%). Based on these promising results, the vision was forwarded that these proteins might serve as a diagnostic target to detect teratoma, possibly by functional imaging such as PET-CT.

Clinically, patient stratification before pcRPLND is paramount in order to minimize overtreatment. Elevated miR371 levels could indicate the presence of viable residual tumor components. However, the clinical implementation of the test in this scenario is currently hampered by the teratoma gap. Further investigations to close this gap are clearly mandatory.

In residual seminomas, the diagnostic accuracy of imaging is likewise modest. Former studies suggested that residual masses > 3 cm contain viable seminoma in a relevant number, while masses < 3 cm mainly consisted of necrosis/fibrosis [74,75]. Accordingly, current guidelines recommend surveillance in residual seminoma < 3 cm [6,32]. However, the management of masses > 3 cm remains controversial. FDG-PET/CT is advocated by the guidelines. If PET is negative, no residual vital seminoma is assumed, and thus, surveillance is recommended. In case of FDG-PET positivity the procedure should be repeated after 6–8 weeks due to its high-risk of false-positive results [76]. If PET remains positive, further therapy should be considered in light of individual characteristics of the case [6]. To overcome this diagnostic inaccuracy, miR371 was investigated in a pilot study of 23 seminoma patients with residual masses after chemotherapy [77]. As control groups, the authors employed 11 patients with complete remission after chemotherapy and 33 healthy males. They found normal miR371 levels after completion of chemotherapy of metastatic seminoma to indicate the absence of viable tumors in residual masses, while elevated levels predicted the presence of vital seminoma. Similarly, Konneh et al. studied serum samples of 15 seminoma patients prior to RPLND, including 10 chemotherapy-naive patients [78]. They reported a specificity of 60% of the test in patients after chemotherapy, while in chemotherapy-naive patients, specificity and sensitivity were 100% and 78%, respectively.

So far, the currently available data on residual seminoma are based on limited patients’ cohorts and are somewhat inconsistent. Therefore, further studies are needed to evaluate the potential advantage of miR371 for evaluation of residual seminoma. In particular, the cutoff of miR-serum levels still needs to be determined.

### 2.7. The Role of miR371 in Current Guideline Recommendations

Guidelines represent the cornerstone for clinical decision-making in the management of TGCTs. They are issued by international or national medical societies with usually annual or biannual updates. The EAU guideline from 2022 envisions an “emerging potential” for miR371 because many studies reported a higher discriminatory capacity compared to the established STMs for diagnosis, treatment monitoring and residual or recurrent viable disease [6]. However, in the view of the EAU, not only are further validations needed, but also technical issues such as laboratory standardization and test availability need to be optimized. The German S3 guideline on TGCT in the 2020 version draws similar conclusions [32], and miR371 is rated as a promising tool that might outperform conventional STMs. Additionally, miR371 seems to correlate with tumor burden and therapy response, but is negative in teratoma or GCNIS. Further validation is needed before the marker can be implemented in routine diagnostics. The 2022 edition of the ESMO guidelines provides a short summary of the key data on miR371, but also points to multi-institutional prospective studies to validate the promising data [79]. The 2019 edition of the American guideline does not refer to the new marker [55].

In all, the international guidelines recognize the huge potential of the novel serum biomarker, but they uniformly request further validation before introduction in clinical practice.

### 2.8. Ongoing Studies

Currently, there are four ongoing studies being registered at clinicaltrials.gov (accessed on 3 November 2022). The prospective multicenter study by the Southwest Oncology Group “A Prospective Observational Cohort Study to Assess miRNA 371 for Outcome Prediction in Patients With Newly Diagnosed Germ Cell Tumors” is evaluating the ability of miR371 to predict the risk of relapse in early stage TGCT patients (cS1–cN0, cM0). The study is scheduled to periodically collect serum samples of 956 TGCT patients within the first three years after diagnosis in order “to estimate the positive predictive value within each of the early-stage testicular seminoma and nonseminoma groups using plasma miRNA 371 expression at relapse to detect germ cell malignancy” (NCI-2019-06177).

The Haukeland University Hospital, Norway, is prospectively conducting the study “MicroRNA-371 as Markers for Disease Activity and as a Tool to Monitor the Effect of Chemotherapy and Early Detection of Recurrence in Patients With Testicular Germ Cell Tumours” (NCT04914026). The aim is to include a total of 350 TGCT patients in all clinical stages and collect blood samples in periodic time intervals. The following four topics are to be addressed: (1) “miR371 as a biomarker in testicular germ cell cancer at orchiectomy”—to predict TGCT prior to orchiectomy at initial diagnosis, (2) “miR371 as a biomarker in testicular germ cell cancer at RPLND”—to identify viable TGCT prior to RPLND, (3) “miR371 as a biomarker in testicular germ cell cancer during chemotherapy treatment”—to monitor of treatment effect, and (4) “miR371 as a biomarker in testicular germ cell cancer and detection of recurrence”—to investigate if miR371 indicates recurrences earlier than conventional STM.

The DRKS00019223 is conducted by the German Testicular Cancer Study Group and aims to document the utility of miR371 for early detection of relapses in patients with cSI on surveillance. The study is scheduled to enroll 250 patients and to follow them prospectively for a period of 3 years.

The ANZUP-led CLIMATE 1906 study is scheduled to enroll 200 patients with GCT cSI and to analyze if postoperatively elevated miR levels represent a risk indicator for relapse [80].

These large prospective studies are of outstanding importance for further validation of the utility of the test and to open the door for implementation of the miR371 in clinical practice.

### 2.9. Certified Test Kit—Practical Handling

Presently, one test kit containing all components for measuring miR371 expression in serum (reverse transcription, preamplification, and qPCR) is commercially available (M371 qPCR test, miRdetect, Bremerhaven, Germany). The test is certified by EU (CE) and can be used for the quantification of the miRNA expression in serum of TGCT patients in every real-time PCR system (e.g., Applied Biosystems, Roche Light Cycler, and Qiagen QIAquant Real-Time Cycler). For the application of this test, blood samples are collected in serum separation tubes (e.g., 4.5 mL Sarstedt, Nümbrecht, Germany) and kept at room temperature for 1 h to allow for complete coagulation after blood aspiration. Then, centrifugation is performed for 10 min at 2000–3000× *g* to separate serum. Aliquots of serum samples can be kept deep-frozen between −20 and −80 °C until processing.

For the measurement of miR371 in clinical laboratories, four analytical steps are required: miRNA isolation, cDNA synthesis (reverse transcription), preamplification, and qPCR. Total RNA isolation is accomplished from 200 μL of serum and is followed by microRNA isolation according to the manufacturer’s instructions (e.g., miRNeasy Mini Kit Qiagen, Hilden, Germany). Automated systems for RNA isolation are also available in many laboratories.

The expression levels of miR371 are calculated as relative quantity (RQ) values in comparison to the endogenous control miR-30b-5p.

The technical staffs in each laboratory are considered to have the expertise to perform and evaluate the M371 test because none of the steps will require extraordinary training. The M371 test can be performed within a day in routine diagnostic laboratories. The development of automated systems for qPCR will reduce the time required for analysis of M371 in the future. Other assays than the one described here are currently in use by renowned research centers. All of these assays are basically similar to each other, but differ in regard to minor technical aspects. The ideal assay to move forward with is yet to be determined.

Whether serum or plasma should be used for measuring miR371 is an unresolved question. The present data indicate that both fluids can safely be employed [81]. However, the normalizer microRNA 30b-5p involves higher Ct values in plasma than in serum. Additionally, it was hypothesized that clotting factors in plasma might partly interact with miR371 molecules, preventing them from detection with the assay [82]. Clearly, both serum and plasma can be used for measuring mir371; however, the measurement results are not directly comparable. In clinical studies, only one of the two methods should be employed.

## 3. Conclusions

The present review focused on the most relevant possible applications of miR371 in the clinical management of TGCTs. In each of the scenarios discussed herein, promising study results are already available. However, further prospective studies are needed before miR371 can be implemented as the new standard STM in daily clinical practice.

## 4. Expert Opinion

The discovery of circulating miRNAs of the miR371-3 cluster as novel STMs in TGCTs represents one of the breakthrough events in the history of this disease. While the introduction of the protein-based tumor markers in the seventies of the last century already significantly changed the clinical management of TGCT, it must be expected that the novel marker miR371 will offer another considerable improvement. This marker fulfils all of the criteria of a valuable tumor marker outlined by Lange and Winfield. The one major drawback of the marker is its lack of sensitivity in teratoma, the so-called teratoma gap. The new marker grossly outperforms the traditional markers, and due to its outstanding sensitivity and specificity, it offers several new applications. Basically, the test may aid clinical decision-making in many clinical scenarios. However, we expect five key applications for miR371: (1) diagnostic work-up of testicular new growths with the potential of sparing frozen section examination in selected cases and obviating the need of surgical exposure in selected very small neoplasms; (2) measurement of miR371 during follow-up involves the potential of early detection of relapses and may permit saving several of the imaging procedures, thus saving costs and energy consumption; (3) the test will aid in the diagnostic work-up of retroperitoneal lymphadenopathies of equivocal size, thus avoiding diagnostic surgery or advanced and expensive imaging techniques; (4) due to its very short half-life, the test will rapidly uncover patients not-responding to chemotherapy, thus aiding decision-making for early change in therapeutic regimens; and (5) the test will help in decision-making regarding postchemotherapy residual masses. Thus, postchemotherapy surgery could be spared in selected cases, although the teratoma gap must be considered. Yet, in residual masses of seminoma, the marker will probably highlight residual vital seminoma, obviating the use of PET/CT examinations.

Currently, an increasing number of research groups is investigating the clinical features of the new marker. Therefore, it may be expected that the promising data will be further consolidated in the near future and that the test will be included in the next editions of international guidelines.

## Figures and Tables

**Figure 1 cancers-15-03944-f001:**
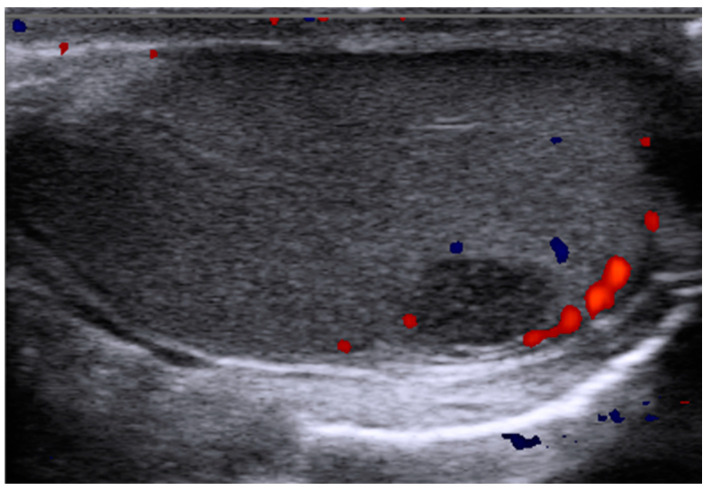
Typical example of an incidentally detected small testicular mass. Sonographic image of left testis showing an 8 mm hypo-echoic mass in the caudal pole region with testicular perfusion visualized by doppler sonography in red and blue. Surgery revealed pure seminoma; miR371 was slightly elevated preoperatively, so inguinal orchiectomy was performed without frozen section examination.

**Figure 2 cancers-15-03944-f002:**
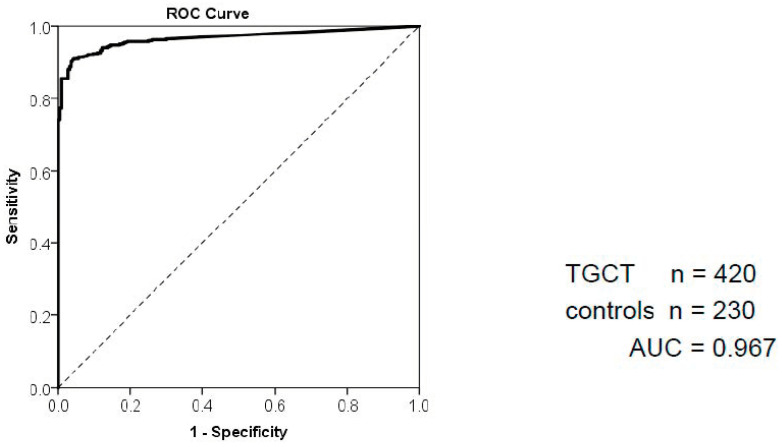
Receiver operating characteristics curve (ROC) showing sensitivity and specificity of miR371 for the primary diagnosis of testicular germ cell tumors (TGCT). The data used in this analysis (420 TGCT patients and 230 healthy controls) are derived from an interim analysis of the study of the German Testicular Cancer Study Group, the final results of which were published later [32]. The area under the curve (AUC) of 0.967 is close to the ideal goal of 1.0. The sensitivity and specificity calculated from these data amount to 89.8% and 93.6%, respectively.

**Figure 3 cancers-15-03944-f003:**
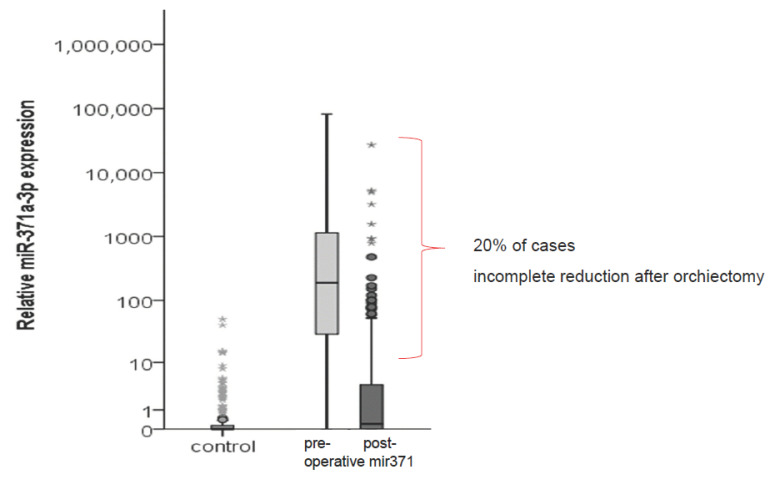
Box and whisker diagram showing the median serum level of miR371 in 230 healthy controls and in 298 clinical stage I (cSI) patients before and after orchiectomy. The data used in this analysis are derived from an interim analysis of the study of the German Testicular Cancer Study Group, the final results of which were published later [34]. There is a significant drop of the median miR level subsequent to orchiectomy; however, 20% of cases do not decrease to normal. This analysis formed the basis for the hypothesis that incomplete decreases after orchiectomy might indicate patients at increased risk of progression. * denotes outliers.

**Figure 4 cancers-15-03944-f004:**
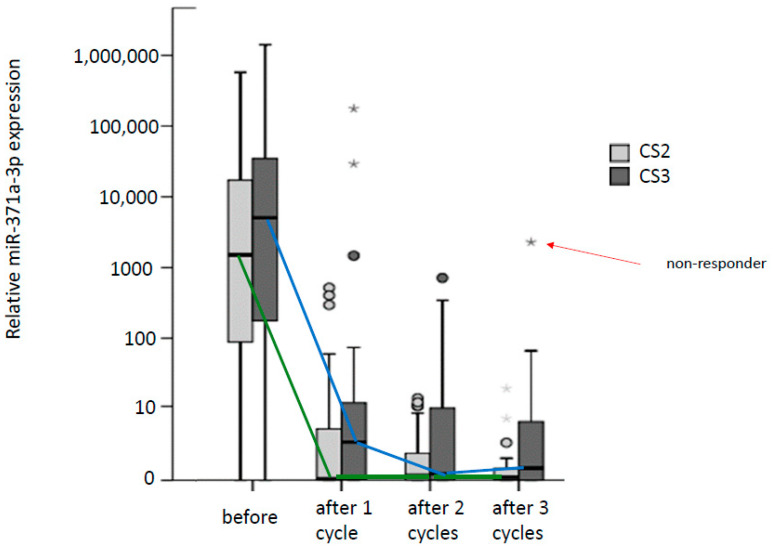
Box plot diagram showing median miR371 levels in patients with clinical stages 2 (n = 69) and cS3 (n = 23) before and after each cycle of chemotherapy. The data used in this analysis are derived from an interim analysis of the study of the German Testicular Cancer Study Group, the final results of which were published later [34]. There is a significant drop in median levels of both cS2 and cS3 patients after the first cycle of chemotherapy. Further decrease is only observed in cS3 patients (blue line), green line denotes cS2 patients. Dots represent outliers. The star in the right upper corner (arrow) denotes one patient not responding to chemotherapy. The graph illustrates that miR levels correlate with the amount of vital tumor bulk.

**Figure 5 cancers-15-03944-f005:**
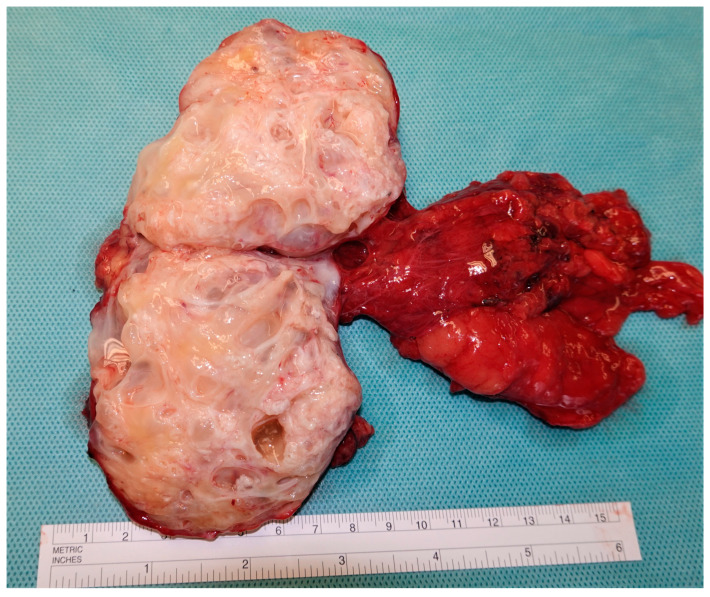
Cross-section of a surgical specimen from the postchemotherapy retroperitoneal lymph node dissection of a residual mass in a patient with metastasized nonseminoma showing the typical macroscopic feature of pure teratoma. This histologic subtype does not express miR371.

**Table 1 cancers-15-03944-t001:** Ideally, blood-based biomarkers should fulfil the following seven criteria by Lange and Winfield [15].

	Criteria
1	The marker is exclusively produced by the malignancy.
2	The marker is secreted into body fluids.
3	The measurement is reproducible.
4	Marker levels in body fluids correlate with actual tumor amount.
5	The marker is detectable in early disease stages.
6	Marker levels correlate with treatment response.
7	The half-life of the marker is short.

**Table 2 cancers-15-03944-t002:** Serum tumor marker miR371 for primary diagnosis of any TGCT and performance characteristics. Prospective studies are indicated with an asterisk.

Author, Year	Number of Patients	Sensitivity (%)	Specificity (%)	AUC	PPV (%)	NPV (%)	Ref.
Dieckmann, 2012	20	70.8	n/a	n/a	n/a	n/a	[26]
Gillis, 2013	80	>90	61	0.89	n/a	n/a	[36]
Syring, 2015	89	84.7	99	0.93	n/a	n/a	[25]
Dieckmann, 2017 *	166	92	84.7	0.94	n/a	n/a	[28]
van Agthoven, 2017	250	89	90	0.95	94	79	[27]
Dieckmann, 2019 *	616	91.8	96.1	0.97	97.2	82.7	[34]
Nappi, 2019 *	110	96	100	0.97	100	98	[37]
Mørup, 2020 *	52	67.5	n/a	n/a	n/a	n/a	[38]
Badia, 2021	69	93.1	100	0.98	100	73.3	[39]
Myklebust, 2021 *	180	89	100	n/a	100	55	[40]
Ye, 2022	51	81.8	100	0.93	n/a	n/a	[41]
Sequeira, 2022	82	93.6	100	0.98	100	96	[42]

Note: AUC = Area under the curve; PPV = positive predictive value; NPV = negative predictive value; Ref. = reference; and n/a = not available.

**Table 3 cancers-15-03944-t003:** Serum tumor marker miR371 for follow up of non-metastasized TGCT patients and its performance characteristics. Prospective studies are indicated with an asterisk.

Author, Year	Number of Patients (Total)	Number of Patients with Relapse	Sensitivity (%)	Specificity (%)	AUC	Ref.
Terbuch, 2018	10	10	n/a	n/a	n/a	[53]
Dieckmann, 2019 *	616	46	82.6	96.1	0.92	[34]
Lobo, 2020	151	34	94.1	n/a	n/a	[48]
Fankhauser, 2022 *	33	10	100	100	1.0	[54]
Nappi 2023	101	35	62.8	100	0.81	[43]

Note: AUC = Area under the curve; Ref. = reference; and n/a = not available.

## Data Availability

The data can be shared up on request.

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
