# Peer review of "MicroRNA-371a-3p—The Novel Serum Biomarker in Testicular Germ Cell Tumors"

_cancers, 2023, doi:10.3390/cancers15153944_

Round 1
Reviewer 1 Report
A comprehensive review for an important new topic.
No major comments.
Reviewer 2 Report
The authors have written a comprehensive review of miR371 in testis cancer.
Some minor improvements can be made:
Section 2.4 is missing data from the SAG group which is consistent with the Princess Margaret findings, as well as presented findings from Lucia Nappi at GU ASCO 2023, later updated at ASCO 2023.
Section 2.4 and 2.5 overlap a lot and it is not clear how they distinguish themselves. Several of the references are similar and some references in 2.4 are missing from 2.3. These sections could be reworked.
Section 2.6 could include the IGR and MSKCC marker decline calculations that prognosticate patients with advanced disease, into favourable decline etc... Authors could postulate that miR-371, with a shorter half-life, might be able to prognosticate patients better in this scenario
Section 2.9 is missing the COG led AGCT1531 study, and the ANZUP led CLIMATE study.
Section 2.10 should elude to other assays in the research field and discuss the fact that the ideal assay to move forward with is yet to be determined.
minor issues only
Reviewer 3 Report
This article represent nice and comprehensive review of the role of mir 371 - 3p in the clinical management of testicular cancer from the group with significant scientific contribution in this field.
Minor comment:
I suggest to add discussion related to differences in plasma vs serum for mir 371-3p detection (Cancers (Basel). 2021 Aug 24;13(17):4260. )
